# Recommendations for the Assessment of Potential Environmental Effects of Genome-Editing Applications in Plants in the EU

**DOI:** 10.3390/plants12091764

**Published:** 2023-04-25

**Authors:** Michael F. Eckerstorfer, Marion Dolezel, Margret Engelhard, Valeria Giovannelli, Marcin Grabowski, Andreas Heissenberger, Matteo Lener, Wolfram Reichenbecher, Samson Simon, Giovanni Staiano, Anne Gabrielle Wüst Saucy, Jan Zünd, Christoph Lüthi

**Affiliations:** 1Umweltbundesamt–Environment Agency Austria (EAA), Landuse and Biosafety Unit, Spittelauer Lände 5, 1090 Vienna, Austria; marion.dolezel@umweltbundesamt.at (M.D.); andreas.heissenberger@umweltbundesamt.at (A.H.); 2Federal Agency for Nature Conservation, Division of Assessment of GMOs/Enforcement of Genetic Engineering Act, Konstantinstr. 110, 53179 Bonn, Germany; margret.engelhard@bfn.de (M.E.); wolfram.reichenbecher@bfn.de (W.R.); samson.simon@bfn.de (S.S.); 3ISPRA (Italian Institute for Environmental Protection and Research), Department for Environmental Monitoring and Protection and for Biodiversity Conservation, Via Vitaliano Brancati, 48, 00144 Rome, Italy; valeria.giovannelli@isprambiente.it (V.G.); matteo.lener@isprambiente.it (M.L.); giovanni.staiano@isprambiente.it (G.S.); 4Ministry of Climate and Environment, Department Nature Conservation, GMO Unit, Wawelska 52/54, 00-922 Warsaw, Poland; marcin.grabowski@klimat.gov.pl; 5Federal Office for the Environment (FOEN), Biotechnology Section, Soil and Biotechnology Division, 3003 Bern, Switzerland; annegabrielle.wuestsaucy@bafu.admin.ch (A.G.W.S.); jan.zuend@bafu.admin.ch (J.Z.); christoph.luethi@bafu.admin.ch (C.L.)

**Keywords:** new genomic techniques, genome-editing, CRISPR/Cas, plant modification, GMO, environmental risk assessment, biosafety regulation

## Abstract

The current initiative of the European Commission (EC) concerning plants produced using certain new genomic techniques, in particular, targeted mutagenesis and cisgenesis, underlines that a high level of protection for human and animal health and the environment needs to be maintained when using such applications. The current EU biosafety regulation framework ensures a high level of protection with a mandatory environmental risk assessment (ERA) of genetically modified (GM) products prior to the authorization of individual GMOs for environmental release or marketing. However, the guidance available from the European Food Safety Authority (EFSA) for conducting such an ERA is not specific enough regarding the techniques under discussion and needs to be further developed to support the policy goals towards ERA, i.e., a case-by-case assessment approach proportionate to the respective risks, currently put forward by the EC. This review identifies important elements for the case-by-case approach for the ERA that need to be taken into account in the framework for a risk-oriented regulatory approach. We also discuss that the comparison of genome-edited plants with plants developed using conventional breeding methods should be conducted at the level of a scientific case-by-case assessment of individual applications rather than at a general, technology-based level. Our considerations aim to support the development of further specific guidance for the ERA of genome-edited plants.

## 1. Introduction

Over the last two decades, technological and scientific progress in biotechnology resulted in the development of a wide range of new genomic techniques (NGTs) to genetically modify organisms and/or to impact the expression of certain genes in the genome of the modified organisms [1]. In particular, the invention of CRISPR/Cas technology led to a boost in the development of different NGT applications in plant breeding and other fields in the last decade [2]. For many countries, this has led to regulatory and policy challenges to accommodate emerging or expected NGT products under their existing biosafety laws and regulations [3,4]. Therefore, some countries decided to review their regulatory framework for biotechnology products, i.e., their GMO legislation, and to propose and implement amendments [5,6]. It is noteworthy that in almost all countries, the level of regulatory oversight for NGT applications, including the requirement of an ERA prior to environmental releases or marketing, is significantly different depending on whether these applications are regulated as GMO or not [4]. The Canadian plants with novel traits (PNT) regulations, which focus on novelty as a regulatory trigger, are probably the most prominent exception in this respect [7]. A number of other countries, including the USA, Argentina, Australia, and Brazil, generally do not regulate genome-edited organisms in a similar manner to GMOs, which are developed with established transgenic approaches, in particular, if no “foreign” transgenic DNA is present in the product [6].

In the EU, the European Court of Justice (ECJ) clarified in 2018 that plants developed with NGTs established using targeted mutagenesis (i.e., genome-edited plants) are subject to Directive 2001/18/EC on the deliberate release and placing on the market of GMOs [8]. The ruling did not address a number of other NGTs, such as cisgenesis, grafting involving GM plant parts (e.g., GM rootstocks), and epigenetic engineering. Prior to these legal proceedings, the competent authorities of the EU member states, the EC, and EU institutions, such as the EC Joint Research Centers (JRC) and the European Food Safety Authority (EFSA), were discussing technical and regulatory matters at the EU level for more than a decade. In 2017, the High Level Group of Scientific Advisors to the EC reviewed the characteristics of new techniques in agricultural biotechnology as a basis for further deliberations [9].

The ECJ decision ended a period of indecisiveness in the EU concerning genome-edited products. However, some challenges regarding the practical implementation of the ruling did remain [10,11]. To address such questions, the Council of the EU requested the EC to conduct a study and—if appropriate, in view of the outcomes of the study—to submit a policy proposal [12]. The EC published the requested study in 2021 [13]. To complement its work, the EC commissioned several technical reports on relevant issues, among other issues relating to the ERA of NGT applications. These additional studies included two reports by the JRC on recent technical developments concerning new techniques in agricultural biotechnology [1] and on current and future market applications of products developed with such techniques [14]. The European Group on Ethics in Science and New Technologies (EGE) provided an opinion on ethical questions associated with the application of genome-edited organisms [15]. The EFSA published several opinions on the ERA of different types of NGT applications, which are highly relevant to the topics discussed in this review [16,17,18]. In another opinion, the EFSA provided an update to its previous opinion on the risk assessment of plants developed with cisgenesis [19,20]. In October 2022, the EFSA published an initial statement describing possible criteria for the risk assessment of genome-edited plants [21].

Based on the outcomes of the EC study and the subsequent discussions with Member States, stakeholders, and the public, the EC published an Inception Impact Assessment (IIA) outlining the objectives and policy options concerning an intended policy initiative on plants obtained using targeted mutagenesis and cisgenesis [22]. Four policy elements are considered for the subsequent development of an impact assessment, which is expected to be available by 7 June 2023 (https://ec.europa.eu/transparency/documents-register/api/files/SEC(2023)2443?ersIds=090166e5f6b76f56 (accessed on 19 April 2023)):Requirements for ERA and approval of NGT products, which on one hand maintain the current high standards of protection and, on the other hand, are proportionate to the risks associated with such applications (i.e., in line with the risk profiles of these applications).Provisions for conducting a sustainability analysis to examine whether and in which way NGT products contribute to sustainability.Appropriate traceability and labelling provisions for NGT products taking into consideration questions related to their implementation and enforcement.Mechanisms for future-proofing the regulatory framework and ensuring the adaptability of the legislation regarding future technological developments.

The EU policy initiative, without any doubt, will have a significant impact on the policies concerning NGT products of non-EU European countries that have special economic relationships with the EU, such as Switzerland and Norway. It will also be met with high interest on a global level, e.g., at the meetings of the parties of the Cartagena Protocol on Biosafety.

Switzerland and Norway are both pursuing work to formulate national policies on NGTs based on their respective national legislation, which are—according to the type of NGT and the scale of cultivation—broadly compatible with the EU approach. The Swiss Parliament has mandated the Federal Council to draft a risk-oriented regulation for certain transgene-free GMOs, which provide an additional value for agriculture, the environment, or consumers, by mid-2024 (see article 37*a*, paragraph 2 of the Swiss Gene Technology Act).

This review specifically focuses on the first policy element, namely, the necessary consideration of safety requirements for NGT products developed using genome editing that would remain a prerequisite for the deliberate release or placing on the market. The discussion provided in this review takes into account the abovementioned, EU-level documents as well as our own work previously published on the subject [23,24,25] and other works available in the literature.

## 2. Challenges Regarding the EC Proposal for a Case-By-Case Risk Assessment of Genome-Edited Plants

The discussion regarding a framework for the future approach towards the ERA of genome-edited plants at the EU level will also be influenced by the opinions and technical reports from the EFSA, which were published starting in 2012 [16,17,18,20,26]. However, a couple of challenges are evident in relation to these opinions and the EC policy documents referring to these studies.

### 2.1. General Conclusions on Risk Assessment versus Appropriate Guidance for a Case-By-Case Risk Assessment

The EFSA issued several opinions on NGT plants produced using cisgenesis/intragenesis and genome editing with site-directed nucleases (SDNs) [16,19,20,26]. Nucleases such as meganucleases, zinc-finger-nucleases (ZFNs), transcription activator-like effector nucleases (TALENs), or CRISPR-Cas-nucleases are commonly called “site-directed nucleases” or “SDN”. We note that a more accurate term would probably be “sequence-directed nucleases” or “sequence-specific nucleases” (as, e.g., in Refs. [27,28]) since they recognize and target specific DNA sequences of a particular length rather than functional genetic elements at specific genomic locations (sites). This is relevant for the discussion of off-target modifications (see Section 3.3 and Section 5). However, the above mentioned EFSA opinions are not stand-alone guidance documents for a case-specific ERA of such applications. Rather, the EFSA assessed whether the available guidance for the ERA of GMOs [29,30] is applicable for the evaluation of food and feed products derived from such NGT plants. Conclusions regarding the suitability of the existing guidance are mostly drawn on a general level, indicating that the principles and the general approach in the existing guidance are indeed applicable to the specific types of NGT plants addressed in the opinions. In addition, the EFSA concludes in a general way that requirements for event-specific data may be different—in some cases lesser—compared with data requirements for established GMOs as assessed by the EFSA over the last years [16,20,26].

The work of the EFSA is summarized by the EC in its study as follows: the “EFSA did not identify new hazards specifically linked to the genomic modification produced via SDN-1 SDN-2 or oligonucleotide-directed mutagenesis (ODM), compared with conventional breeding and techniques introducing new genetic material” [13]. For explanation: SDN-1 applications exploit the non-homologous end-joining (NHEJ) repair pathway to introduce random mutations (substitutions, insertions and deletions) at a double-strand break site in the plant genome [16]), whereas the SDN-2 approach makes use of a template DNA to generate an intended sequence modification at a double-strand break site in the plant genome via the homology-directed repair (HDR) pathway in plants. In contrast, SDN-3 applications exploit both NHEJ and HDR pathways to insert longer, recombinant DNA constructs at a targeted genomic location [16]). The ODM approach uses oligonucleotides for the introduction of small targeted mutations in the plant genome (one or a few adjacent nucleotides in length). The genetic changes that can be obtained using ODM include substitutions, insertions or deletions [9]).

The above mentioned conclusion is only valid if it refers to the general observation that no additional new risk issues other than those addressed in the EFSA guidance document for the ERA of GM plants [29] may be expected. However, the scope of the risk issues in the ERA guidance is very comprehensive and was developed with the aim to address all potential hazards that may be associated with established GMOs and a variety of different GM traits. Thus, the scope includes any specific biological hazards posed by emerging genome-edited plants.

### 2.2. General Comparability of Genome-Edited Plants with Conventionally Bred Plants

In their study on NGTs, the EC states: “As concluded by EFSA, similar products with similar risk profiles can be obtained with conventional breeding techniques, certain genome-editing techniques and cisgenesis” [13]. This conclusion is based on the analysis by the EFSA, stating that on-target, as well as off-target mutations introduced with genome editing using SDN methods, are of the same type as mutations occurring in conventionally bred plants, including spontaneous mutations and those induced with physical and chemical mutagenesis [16]. The notion of general comparability of genome-edited and conventional plants in this respect is a cornerstone of the conclusions by the EFSA and the EC concerning the potential risks of NGTs and, in particular, genome-edited plants. However, and as discussed further in Section 3, recent scientific findings indicate that spontaneous mutations are not distributed randomly throughout the plant genome, as assumed previously, but occur at a higher frequency in intergenic regions of the genome [31]. Genome editing using SDN methods, on the other hand, is capable of introducing mutations in functionally important genome regions that are “protected” to some extent from mutations induced with conventional techniques [32]; this is one of the reasons why genome editing is regarded to be a very powerful technique. However, it also implies that there are relevant differences between mutations induced with genome editing as compared to sequence changes introduced using conventional breeding techniques.

### 2.3. Selective Use of the EFSA Opinions to Conclude on the Safety/Risks of Genome-Edited Plants

Not all the available EFSA opinions addressing the ERA of genome-edited plants are considered equally by the EFSA and the EC for drawing conclusions. The EFSA did not address all SDN-1/SDN-2 applications in a single opinion. The opinion of Naegeli et al. [16] is explicitly focused on genome-editing applications. However, it does not address the full range of all possible genome-editing applications. A separate EFSA opinion on plants obtained using synthetic biology addresses complex genetic modifications created with SDN-1 techniques [17]. Such complex modifications may be created using genome-editing tools such as CRISPR/Cas, which are designed for multiplexing, i.e., introducing multiple changes in particular genomic loci or editing multiple genes simultaneously [33].

The conclusions of the “SDN-1/2” opinion [16] are very general, as outlined above. However, the conclusions drawn in the latter opinion [17] concerning a case study on a genome-edited plant containing complex genetic modifications (a low-gluten wheat plant produced with targeted mutations of multiple alpha-gliadin genes using a CRISPR-Cas9 SDN-1 approach) are more specific. It is stated that “the large number of mutations required to achieve gluten-free wheat is far beyond any plant previously assessed” by the EFSA [17]. They would require a comprehensive risk assessment approach based on the existing EFSA guidance for an ERA [29] and a food/feed safety assessment [30]. Nevertheless, the opinion indicates that it would be challenging to obtain a robust comparative safety assessment for genome-edited plants with complex metabolic modifications, such as the genome-edited wheat assessed in the case study [17]. However, different to the conclusions of the SDN-1/2 opinion, the latter opinion is neither referenced nor discussed explicitly in the overview analysis provided by the EFSA [18], the EC study [13], or the IIA [22].

In effect, this approach followed by the EFSA led to a situation where multiplexed genome-editing applications with complex physiological modifications were not taken into account sufficiently for drawing conclusions by the EFSA or by the EC. As almost half of the studies addressing genome-editing SDN-1 applications, which are regarded relevant for future agricultural use [34], consider products with complex genetic modifications [35], the approach chosen by the EFSA and the EC does not properly reflect the whole spectrum of genome-edited plants. The conclusions contained in Naegeli et al. [17] concerning these genome-editing applications, therefore, are highly relevant regarding the overall picture. For the first time, the EFSA statement concerning a proposal for criteria for the risk assessment of plants produced with targeted mutagenesis, cisgenesis, and intragenesis [21] addressed the issues of complexly modified genome-edited plants. However, it remains to be seen whether this amounts to a change in the overall approach of the EFSA and the EC.

### 2.4. Generalized Conclusions Regarding the Detection and Identification of Genome-Edited Plants

According to the current regulatory system in the EU, issues concerning the detection and identification of GMOs are not addressed within the remit for risk assessment. However, it is important to note that both the EC study on NGTs and the IIA highlight that there are challenges to implementing and enforcing the current GMO legislation in relation to its traceability and labelling requirements [13]. These concerns are based on the finding that in some cases of plants produced with targeted mutagenesis or cisgenesis, analytical methods might be capable of detecting the product, but might not be capable of identifying the technique used to obtain a specific sequence change. These concerns were previously stated in a report by the European Network of GMO Laboratories [36]; the respective challenges are also summarized in Grohmann et al. [37]. Again, the conclusions on these challenges to detect and identify NGT products are presented in a very general way.

The EC study, however, does not address a number of important points: (i) The availability of appropriate molecular information is a key issue for the successful detection and identification of both established GMOs and NGT applications. A lack of appropriate molecular information poses significant challenges for the development of screening and detection/identification methods for GM plants, similarly as it does for ERA. This is particularly relevant for GMOs that are not authorized or notified under EU legislation yet and/or that do not contain commonly used transgenic elements [38,39]. (ii) The development of methods for the analytical detection of both new GMOs and genome-edited plants is a dynamic process. Based on sufficient molecular information for genome-edited products, new methods are and will be developed, which may eventually offer the same reliability to detect and identify emerging genome-edited products as the analytical detection methods available for established GMOs [40,41]. (iii) The complexity of many modified genome-edited products (45% of SDN-1, according to Kawall [35]) may simplify the development of analytical methods for the detection and identification of the respective products. Genome-edited plants with multiple- and/or larger-sized genetic modifications are less difficult to detect and identify than genome-edited plants with a single, minute genetic sequence change. (iv) Identification of any GMO, including transgenic products, is generally based on a high probability that the detected DNA sequence(s) may be considered unique and would not likely occur in other products or natural varieties.

## 3. Comparability of Genome-Edited and Conventionally Bred Plants

The EC study on NGTs [13] and the related EFSA opinions [16,18] compare genome-edited plants with plants developed using conventional breeding approaches, including classical mutagenesis, at a general level. We argue that such generalized comparisons concerning large groups of different plants containing widely different traits are seriously flawed from a risk assessment perspective. For the ERA, a specific assessment of the characteristics of the individual plants, their newly established traits, and their interactions with the receiving environment cannot be replaced with general “theoretical” considerations. Our arguments in this respect address the following issues:Theoretical assumption of the “likeness” of mutations introduced with different techniques.Consideration of the depth of intervention, i.e., the complexity in the resulting phenotypic outcomes.Consideration of the difference in the occurrence of unintended genetic modifications.Consideration of the higher speed of development of genome-edited plants.

### 3.1. “Likeness” of Mutations Introduced by Different Techniques

Firstly, assumptions that the mutations introduced with spontaneous natural processes and classical mutagenesis are “similar” to the ones introduced with genome editing are not scientifically sound. Thus, an assumption of general “likeness” may be misleading.

Conventional breeding methods including classical mutagenesis—at a general theoretical level—can result in a very broad range of different types of mutational changes. These include point mutations, insertions, and deletions of sequences (indels) as well as larger-scale chromosomal aberrations. Therefore, any outcome of SDN-based, genome-editing approaches may be compared with conventionally introduced types of mutations at a general level. However, such theoretical comparisons do not take into account whether it would actually be feasible to develop plant varieties corresponding to particular genome-edited plants with conventional approaches or whether such conventionally bred varieties are indeed used in agricultural practice. In our opinion, the theoretical argument is thus not particularly helpful for risk-oriented considerations. We note that the Canadian PNT regulations, since their establishment in 1996, contain provisions that newly developed products are only considered “not novel” if plant varieties with similar traits are actually in practical use in current agriculture [42].

Secondly, an earlier review indicated that the theoretical comparison of the range of mutations generated with conventional and genome-editing methods is flawed: conventional mutagenesis does not generate a random distribution of mutational events across the genome of a plant, whereas genome-editing methods allow the introduction of mutations into parts of the genome that are somewhat protected against spontaneous genetic alteration [32]. A study conducted by Monroe et al. [31] using *Arabidopsis thaliana* provides further compelling evidence that natural spontaneous mutations occur in a biased way across the genome, with a clear preference for non-functional regions in plant genomes. Their work indicates that mutation rates in the model plant *A. thaliana* are lower in genomic regions that are functionally more important and where mutations are more frequently harmful [43]. Monroe et al. [31] found that mutations occurred at significantly lower rates in actively transcribed genomic sequences, i.e., at a 58% lower rate in gene bodies relative to flanking intergenic regions and a 37% lower rate in essential genes relative to non-essential genes. Thus, mutation rates in actively expressed genes with important functions are reduced by two-thirds in comparison to the mutation rates found in intergenic sequences.

These significantly different mutation rates at different genomic loci are due to intrinsic characteristics of the respective genome regions rather than to subsequent events, i.e., natural selection [44]. The intrinsic bias of mutation frequencies at different genomic loci is related to epigenetic features known to affect the locus-specific level of DNA repair and, thus, the vulnerability of specific genome loci to damage [31]. It is evident that mutations introduced with SDN tools for genome editing would not be subject to such biases, since the presence of the SDNs throughout the editing process will result in cuts at SDN target loci that were repaired to match the original genomic sequence until a mutation is eventually introduced. This new evidence is strengthening earlier conclusions that applications of genome editing using SDNs can result in new genetic combinations that would not likely occur naturally or with conventional breeding [32].

Another argument put forward to compare genome-edited plants with conventional plant varieties is that most genome-edited plants do not contain integrated recombinant sequences that include “foreign” DNA, i.e., DNA sequences obtained from other organisms than the modified plant species [6]. As noted by Entine et al. [6], this criterion, however, bears no relationship with the presence or absence of a novel hazard or any specific risk. Similarly, Gould et al. [45] argued that the size and source of the genetic material inserted into genome-edited plants should not be the most important factors in relation to testing requirements for risk assessment.

### 3.2. Depth of Intervention and Possible Complex Modifications

Another important difference between conventional breeding approaches and genome-editing-based techniques is the level of multiple genetic changes that may be introduced with the respective approaches. While the number of simultaneous changes that can be possibly introduced with classical mutagenesis and crossbreeding is limited due to the necessary crossbreeding steps, multiplexing of genome-editing-based SDN techniques is routinely performed [46]. Thus, the complexity of genetic change that may be achieved with the latter approach is significantly higher [35]. This, in turn, can facilitate a broader range of phenotypic changes and a higher depth of simultaneous interventions in comparison to conventional approaches, which require multiple breeding cycles. Complex interventions also frequently result in traits that are considered “novel”, i.e., traits that are not present in natural populations and/or used agronomically [47].

A higher depth of intervention at the genetic and/or phenotypic level, however, is posing more challenges for a regulatory risk assessment [17]. The EFSA statement from October 2022 indicates that the current assessment approach for GMOs may not be feasible for plants with a high number of inserted or modified sequences [21]. Currently, the risk assessment starts with the assessment of single modifications and subsequently assesses multiple, stacked modifications. The statement also highlights that a comparative analysis may not always be feasible for genome-edited plants with complex traits for which a comparator cannot be identified easily [21]. Since a significant proportion of genome-edited plants developed with SDN-1-based approaches is regarded to contain complex genetic and/or phenotypic alterations, substantial risk assessment challenges will likely arise in more than 45% of all agriculturally interesting plants modified with SDN-1 techniques [35].

### 3.3. Difference in the Occurrence of Unintended Genetic Modifications

In their 2020 opinion, the EFSA stated that genome editing will result in fewer unintended genetic modifications compared to certain conventional breeding techniques, such as classical mutagenesis [16]. However, this does not take into consideration the removal of unintended modifications during subsequent breeding steps that are inherently necessary for conventional breeding schemes. Additionally, the effects of the in vitro steps necessary to express genome-editing tools in the target cells and the different tendencies of the existing methods for genome editing to induce unintended modifications [48] are disregarded.

Of importance for drawing a comparison is also the fact that unintended modifications will likely occur at different frequencies and at different genomic locations using genome-editing or conventional techniques, respectively. Modifications due to classical mutagenesis are thought to be induced randomly throughout the genome and—at lower mutational rates—would be subject to the same bias of distribution in the genome as detected for spontaneous mutations (see Section 3.1). In contrast, off-target modifications introduced with genome-editing tools are occurring predominantly at genomic loci sharing sequence homologies with the intended target sites [49]. Thus, the frequencies of off-target modifications will be considerably higher at genomic locations sharing functional similarities with the targeted genetic sequence. Concerning the final number of unintended modifications present in a marketable plant variety, the effects of breeding steps following the genetic modification need to be considered as well. The probability that unintended modifications—in particular those, which are not genetically linked to the intended trait—are removed is increasing with the number of subsequent backcrossing steps. Classical mutagenesis typically involves a significant number of backcrossing steps to ideally retain only the intended trait(s) and remove any unintended mutations. Sidestepping or considerably curtailing such a backcrossing regime, e.g., when fast-track genome editing is conducted directly in elite germplasm [13,50], will reduce the margin of safety inherent to the approach used for classical mutagenesis. The number of mutations initially present in the mutagenized plants may be higher for classical mutagenesis compared to genome editing—as highlighted by the EFSA [16] and the EC [13]. However, this difference may not be reflected similarly in the number of unintended modifications present in the final breeding products.

### 3.4. Higher Technical Speed of Genome Editing in Plant Development

Genome-editing-based approaches are typically credited with a higher speed of development of plant products intended for agricultural application by scientists who are actively developing such products [51]. Compared to developments in plants using conventional cross-breeding or mutation breeding, or to the development of transgenic plants, which on average take about 8–12 years, respectively, the technical development of plants with targeted mutagenesis using genome editing is expected to be significantly faster. Such developments on average are assumed to take only 2–5 years, due to the ease in application of the available genome-editing tools, the specific targeting of the approach, and the fact that few or no backcrossing steps are required [51]. Due to the latter aspect, the direct modification of elite lines and commercially useable varieties is possible [50]. Similarly, perennial plants with longer generation cycles as well as vegetatively propagated varieties may be modified more readily with genome editing than with conventional breeding approaches [23]. If fewer crossbreeding steps are conducted to shorten the time of development, then the length of the observational period until the breeding products reach the agricultural market and the possibilities to spot and remove any unintended and potentially adverse effects associated with the developed plants and their application are decreased.

It should be noted that the technical development of a genome-edited plant is just one step in the development of a market-ready plant product. The lower-than-expected speed of commercialisation for genome-edited plants and the few products available on the market indicate that currently, other factors than technical development do play a role.

## 4. Considerations for the Assessment of Trait-Related Effects in Genome-Edited Plants

The EC Inception Impact assessment pointed out that the ERA for NGT plants, including plants modified with genome editing, needs to take into account the specific technique used for modification, the type of modification, and the novelty of the trait [13]. As indicated in earlier work, such an ERA has to address adverse effects resulting from intended and unintended modifications present in the NGT plant, in particular, effects that are due to the characteristics of the modified plant species and its interaction with the receiving environment and its use in agriculture [9,23]. The EFSA indicated that data requirements for the ERA of genome-edited plants will, however, mainly depend on the modified traits present in the assessed plants [16].

Because newly developed traits are a focus of the ERA, the following sections will examine the range of different traits that may be generated with NGT and particularly with genome-editing methods. It will also address the challenges associated with certain types of traits present in NGT plants that are currently developed for use in agriculture, including a brief analysis of a number of examples of different genome-edited plants.

### 4.1. Trait Categories Developed Currently in Genome-Edited Plants

Substantial work has been invested recently to identify which NGT and genome-editing applications could be expected to be introduced into agricultural markets in the foreseeable future [14,25,34,49,52]. The EC study predicts that a broad range of NGT plants may be available in the near future, including plants that are more resistant to diseases, adverse environmental conditions, and climate change effects in general and that carry improved agronomic or nutritional traits, which can be grown using fewer agricultural inputs, e.g., plant protection products, and which are developed with faster plant breeding [13].

The abovementioned publications address slightly different but overlapping ranges of different NGT applications, with a common focus on all types of SDN applications established with CRISPR/Cas and other genome-editing nucleases.

### 4.2. Simple versus Complex Traits Developed with Genome Editing

As indicated above, some of those traits may be introduced with “simple” genomic modifications, while many complex and/or novel traits—such as traits resulting in resistance to biotic and abiotic environmental stressors and adaptations to climate change—may require complex genomic alterations involving multiple alleles, multiple gene copies, or multiple different genes [35,49].

Examples of “simple” modifications are particularly numerous in applications targeting herbicide resistance, which may be facilitated by single dominant alleles [25]. In SDN-1 applications to facilitate breeding and herbicide resistance, single-gene knockouts are overrepresented [35].

Complex genomic modifications, i.e., the simultaneous modification of multiple gene copies (alleles, members of a gene family) and of multiple different genes (using multiplexing), are common in SDN-1 applications for industrial purposes. In other trait categories of SDN-1 applications, i.e., biotic or abiotic stress resistance, agronomic value, and food and feed quality traits, developments based on either single-gene knockouts or complex genomic modifications can both be found in significant numbers [35], as indicated in Table 1.

Therefore, a substantial proportion of forthcoming genome-editing applications are complex developments at the genomic and/or phenotypic level. Important trait categories such as traits targeting agronomic value (34% of analysed SDN-1 applications) as well as food and feed quality (26% of analysed SDN-1 applications) include a significant fraction of complex genomic modifications. This is particularly true for developments targeting traits such as biotic and abiotic stress resistance, where approximately half of the applications are based on complex genomic modifications. In addition, some single-gene knockout modifications in those trait categories can also act as developmental regulators and thus exert complex effects on morphology, development, and reproduction, leading to complex, sometimes unpredictable, physiological effects. A significant number of applications will furthermore involve plant species that are less commonly used in current agriculture and traits that are “novel” in comparison to existing agricultural varieties [23].

Due to the range of different traits developed in genome-edited plants, a number of different risk issues needs to be addressed during an ERA. It is noteworthy that the specific risk issues triggered by a genome-edited plant will significantly differ between the particular plant x trait x intended usage combination. The latter aspect is important because it determines the interaction with and the exposure of the respective receiving environment for a particular genome-edited plant. As discussed earlier, for some genome-edited plants with resistances to different plant pathogens, the intended modifications may lead to unintended pleiotropic effects on morphology, development, physiology, or composition of the modified plants with relevant implications for ERAs [52]. Thus, a truly case-specific approach to risk assessment based on a robust problem formulation according to the ERA guidance document of the EFSA Panel on Genetically Modified Organisms [29] is required to sufficiently assess the particular risk issues relevant for specific genome-edited plants. Since the level of knowledge and familiarity concerning relevant ERA aspects for genome-edited plants carrying complex genomic modifications and/or novel traits is limited, significant challenges for conducting an appropriate ERA can be expected [53]. Some of these challenges are outlined in the following sections.

### 4.3. Risk Considerations for Different Genome-Edited Plants with Traits from Different Trait Categories

Some examples for genome-edited plants harbouring different traits are discussed in the following sections. The examples were chosen to represent different categories of traits and genetic modifications of varying complexity as outlined in Section 4.2. The examples comprise different plant species, including annual crops and perennial plants, respectively. Included are genome-editing applications at different stages of development or market introduction (marketed or field-tested outside the EU, advanced stage of research and development, early research phases). The examples chosen for analysis also reflect different risk scenarios that may arise for genome-edited plants.

In this review, the following examples of traits and genome-edited plants are addressed:Herbicide resistance (HR) (different genome-edited HR crops, Section 4.3.1).Disease resistance (genome-edited apple trees, Section 4.3.2).Altered composition (genome-edited wheat, Section 4.3.3).De novo domestication (genome-edited tomato, Section 4.3.4).

The latter two examples are applications which cannot be generated easily with other breeding approaches, such as conventional breeding schemes or established genetic modification techniques, i.e., transgenesis.

#### 4.3.1. Genome-Edited HR Crop Plants

Currently, weed management with herbicides is a cornerstone of high-output commercial agriculture. Since the 1990s, HR crops that allow for weed management with broad-spectrum, post-emergence herbicides were developed using biotechnology [54]. Genome-editing methods are also used to introduce targeted genetic modifications into endogenous genes in a range of commercially relevant, agricultural plants to develop HR varieties [55]. Such HR traits are introduced in major crop plants including oilseed rape, maize, rice, wheat, soybean, cassava, and potato as well as in tomato, linseed, peppers, and watermelons [56]. A variety of genome-editing methods are used to create these plants, including SDN-1/2/3 and ODM techniques, base, and prime editing. The developed HR traits confer resistance against several herbicidal substances such as glyphosate, the ACCase inhibitors aryloxyphenoxy propionate and haloxyfop-derivates, and a number of ALS-inhibitor herbicides such as imazamox, imidazolinone, imazethapyr, chlorsulfuron, and sulfonylurea [56,57]. According to the JRC, one genome-edited HR oilseed rape variety is commercially marketed outside the EU and another ten varieties are in advanced research and development or pre-commercial phases [14].

Genome-edited and transgenic HR plants trigger similar risk issues: The focal issues for the ERA of such traits are indirect effects on biodiversity as well as ecological and health effects resulting from the change in agricultural practice towards the increased use of broad-spectrum herbicides. Additionally, the spread of HR volunteer plants in and outside of cultivated areas, the flow of HR genes to related (weedy) species through hybridization, and the development of HR weeds upon long-term, regular use of the respective herbicides are relevant concerns [54]. During cultivation of ALS-inhibitor HR rice in Italy and the USA, hybridization with related wild species occurred, and HR-resistant weeds emerged [58]. Recent results from ecotoxicological testing indicate that direct effects of glyphosate-based herbicides on biodiversity might have been overlooked and should be considered during ERAs [59].

The possible occurrence of unintended, adverse, off-target modifications that are genetically linked to the HR traits should also be assessed. The possibility that the elevated levels of modified EPSPS proteins in glyphosate resistant plants could lead to pleiotropic effects, such as elevated auxin content and increased fecundity, as identified in *Arabidopsis*, needs to be taken into account [60].

We agree with the EC that the use of HR genome-edited crops does not contribute to the aim of increasing the sustainability of current conventional farming practices in the EU [13]. Overall, we support the notion stated by Hussain et al. [57] that the possible environmental risks of these genome-edited HR crops must be considered prior to cultivation and commercialization.

#### 4.3.2. Disease-Resistant (DR), Genome-Edited Apple Trees

Plants with resistance against diseases are an important breeding objective, since plant pathogens are causing substantial crop and food losses and are significant factors impacting food production and agricultural sustainability. However, varieties with stable and effective resistance traits against agriculturally important pathogens are difficult to develop [61]. The use of genome-editing methods to facilitate and speed up the development of DR plant varieties is addressed in several recent reviews, focusing mainly on DR genome-editing applications in major crop or vegetable species, such as wheat, rice, maize, soybean, oilseed rape, tomato, and potatoes [62,63]. In addition, other plant species including perennials, such as grapevine, citrus and apple trees, etc., are targeted [62].

In the apple, genome editing was used to knockout a host receptor gene that is essential for the development of apple fire blight disease caused by the bacterial pathogen *Erwinia amylovora* [64]. With regard to the ERA of the genome-edited apple, several aspects are relevant, which are also common issues of assessment addressed for GM apple trees. These include the accidental dispersal of the genome-edited apple via pollen, seeds, or root-suckers as well as the outcrossing into other apple trees, particularly into wild apple species with conservation value. Intensification of apple cultivation due to the availability of fire blight-resistant commercial varieties could indirectly affect local pollinator populations, such as wild bees and others, due to the increased use of commercial bee and bumblebee pollinators [65]. 

Important considerations are the possible effects on the target organism: On the one hand, this concerns the emergence of *Erwinia* strains that overcome the resistance trait present in the DR apple and would then be able to threaten the genome-edited apple as well as other non-GM apple varieties harbouring similar DR traits. For a different disease, apple scab, exposure of the pathogens to a DR, GM apple line led to the development of more aggressive pathogen strains within seven to eight years [61,66]. On the other hand, different pathogens could occupy the niche of *E. amylovora*, leading to a change in the pathogen spectrum and infestation status of apple trees, necessitating additional pest management using pesticides or antibiotics. Such incidents may happen with GM and non-GM apples; thus, they should also be addressed during the ERA of DR, genome-edited apple trees.

In general, apple trees, including the particular genome-edited apple, are posing challenges for risk assessment and risk management since they are not only cultivated in commercial plantations or orchards but also in private gardens and public spaces in cities and villages. Feral genome-edited apple trees could spread naturally or with human involvement. Their long lifecycle is also posing challenges for the assessment of long-term effects in comparison with annual GM crops.

#### 4.3.3. Genome-Edited Wheat with Low Gluten Content

Many recent applications of genome editing in plants address breeding goals related to food and feed quality [55]. The developments in this field do not only target major crop species but a wide variety of agriculturally relevant plant species, including vegetables and fruit plants [67]. The pursued goals are manifold, e.g. modifications to improve nutritional quality, flavour, texture, shelf-life, and post-harvest pathogen resistance. Genome-edited wheat with a lower gluten content is a prominent example for these developments and is aimed to alleviate nutritional side-effects caused by the current levels of gluten in wheat [68].

The common approach in all of the mentioned applications is to modify metabolic pathways by knocking out or modifying genes that encode enzymes or regulatory elements in biosynthetic pathways [55]. In some cases, only a single target gene and a limited number of alleles have to be modified, e.g., to remove anti-nutrients [69] or change the composition of a specific substance class, such as starch components [70,71] or fatty acid composition [72]. In other cases, e.g., to reduce the gluten (α-gliadin) content in wheat, a complex, highly multiplexed approach is necessary to modify the alleles of several genes [68]. As example from another plant species, complex modifications were introduced into tomato plants to increase the lycopene content in their fruits [73].

Both types of applications either targeting single genes or multiple alleles of different genes, respectively, can pose challenges for an ERA, as demonstrated by the following two examples: The ecological consequences of the genome-edited *Camelina sativa* variety with modified fatty acid content due to less complex modification were analysed by Kawall [74]. The EFSA assessed the low-gluten, genome-edited wheat as a case study for genome edited products with highly complex modifications. The analysis focused on aspects of molecular characterization and ERA [17].

The EFSA noted that the low-gluten wheat was developed by firstly integrating the transgenic CRISPR/Cas components into the genome of the parental wheat line—a process which may lead to unintended changes in the genome. The respective transgenic construct was removed with segregation after up to 35 of the 45 different α-gliadin genes were modified using CRISPR/Cas in one of the GM wheat lines [68]. Due to the non-directed process of SDN-1 mutagenesis, different indels were created at the modified target sites, i.e., either insertions of ectopic DNA sequences or deletions of genomic sequences of different length. These modifications resulted in a reduction in the adverse immunoreactivity against α-gliadin epitopes by 85% in comparison with wild type wheat. This was caused by the deletion of immunodominant epitopes and/or by a reduced expression of modified α-gliadin genes. We support the EFSA’s conclusion that this low-gluten wheat is highly challenging with respect to molecular characterization of the product: On the one hand, the absence of transgenic sequences as well as of unintended off-target edits needs to be assessed. On the other hand, a characterization of any specific modifications at all modified loci is necessary to assess whether and which modified proteins may be expressed in the genome-edited wheat. Furthermore, the genetic stability of the modifications needs to be assessed [17].

Regarding ERA and food safety, the allergenic, toxic, and anti-nutritional properties of newly expressed modified α-gliadins need to be assessed; which is a challenging task for a genome-edited plant containing a large number of different mutations such as low-gluten wheat. The EFSA concluded that the complexity in this SDN-1 application is far beyond any GM plant that was assessed previously [17]. However, the EFSA’s conclusions also underline that, in general, the current ERA approach for GMOs is adequate and sufficient for such genome-edited plants.

#### 4.3.4. De Novo-Domesticated, Genome-Edited Tomato

De novo domestication, e.g., as described by Li et al. [75] and Zsögön et al. [76] for tomatoes, represent a recent development to speed up plant breeding at an exceptional pace [77]. The approach is based on the editing of key regulatory genes to rapidly develop plant varieties with features resembling domesticated crops from wild forms while retaining desired properties such as strong resistance toward pathogens or salt tolerance [78]. In the above-cited examples, characteristics associated with domesticated tomato plants were established in different lines of *Solanum pimpinellifolium* with the simultaneous modification of four or six genomic loci, respectively. Due to the genetic changes introduced with genome editing, the plants had an increased fruit number and the size, shape, and nutrient content of the fruits were comparable with domesticated tomato plants. In addition, the plant architecture and growth characteristics resembled currently cultivated tomato. This approach is seen as a way to directly develop new crop varieties from wild plants in order to exploit their genetic diversity. Due to the availability of newly developed genome-editing tools, the approach is considered a fast and technically feasible alternative to classic breeding programs [79]. De novo domestication is also discussed as a possible approach to develop climate change-resilient crops from wild relatives that are more tolerant to abiotic stress [80].

However, such de novo-domesticated plants are novel products without any “history of safe use” due to their wild type genetic backgrounds, particularly regarding food and feed safety. They thus require a comprehensive and robust risk assessment to detect any potential hazards associated with their agricultural as well as food and feed use. The current approaches used for the ERA of GM crops will need significant improvements to assess the phenotypic and nutritional characteristics of de novo-domesticated plants appropriately in order to maintain the current level of food and feed safety.

## 5. Considerations Concerning the Assessment of Unintended Genetic Modifications and Their Consequences on a Case-By-Case Basis

The EC study on NGTs [13] acknowledged that the application of genome-editing methods can lead to unintended genetic modifications in the modified plants. As indicated in Section 3.3, the EFSA concluded that the analysis of potential off-target edits would be of very limited value for the risk assessment of plants established with SDN-1/2 applications [16]. We note that this is neither in line with current evidence addressing unintended modifications nor with advice from risk assessment experts from other countries [81].

The efficacy of CRISPR-Cas9 systems for genome editing is typically correlated with their level of off-target modifications [82]: increased efficacy tends to come at the price of higher off-target mutation rates [82], i.e., there is a trade-off between efficacy and targeting precision, and the number of off-target modifications depends on the design and execution of the particular genome-editing approach. Not all genome-editing approaches are optimized for maximum targeting precision and thus a very low-level of off-target modifications. On the contrary, some applications intentionally use genome-editing tools with a lower level of specificity/precision, e.g., to edit a number of slightly different target sequences with appropriate efficiency [23].

In addition, the current evidence base regarding quantification and assessment of unintended modifications is still very limited. A recent meta-analysis of studies addressing off-target modification highlights this fact [83]. The systematic review conducted by Sturme et al. [83] analysed 107 studies addressing the occurrence of off-target modification caused by CRISPR SDN-systems in different plant species. However, only eight of these studies used whole genome sequencing (WGS) methods and only five of them used an untargeted WGS approach. Most of the other studies used targeted PCR-based methods that detect off-target modifications only at a few preselected genomic sites (mostly fewer than 10 loci). Since two studies based on an unbiased WGS analysis identified off-target modifications (indels and base substitutions) at loci that were not predicted with bioinformatics tools, we believe that the evidence gathered from targeted and thus biased assessments is not sufficiently conclusive.

Another issue indicated by Sturme et al. [83] is that the screening for off-target modifications is mostly performed in the early research phases to establish the specificity of the guide RNAs used for a particular goal, rather than the specificity of the method or nuclease in general. Thus, the collected data are not appropriate to address whether off-target modifications are actually removed during subsequent breeding steps, as assumed by the EFSA [16].

Other relevant types of unintended modifications are insertions from vector backbone sequences used to introduce the genome-editing tools or unremoved transgenic insertions of CRISPR/Cas expression cassettes. Sturme et al. [83] indicate that the insertion of large vector-derived sequences in the respective target site was observed in one study [84]. Another study [85] assessed the occurrence of insertions of vector backbone sequences in the genome of T_0_ plants with WGS and found such insertions at five different genomic locations. Thus, the possible presence of such foreign DNA insertions is a plausible concern and should be appropriately assessed, as indicated by Lema [81]. Unfortunately, the assessment of unintended modifications is not addressed at all in the recent EFSA statement on the risk assessment of plants produced with targeted mutagenesis, cisgenesis, and intragenesis [21].

As a way forward, we suggest that our earlier recommendations for a 10 step workflow for a structured, case-specific assessment of unintended modifications should be implemented in the risk assessment of genome-edited plants [23,25].

## 6. Discussion

As indicated in Section 2 and Section 3, the EC [13] and EFSA [16,18] conclude on several relevant issues which are important for risk assessment only in a very general way. This includes general conclusions on the possible risks associated with whole groups of genome-editing applications and the comparability (“likeness”) of genome-editing applications with conventionally bred plants as well as other regulatory questions, e.g., the possibilities to detect and trace genome-edited plants.

Firstly, we note that such a general approach is not in line with another overall conclusion by the EC and EFSA, namely that the case-by-case approach, which is one of the general principles for the ERA and enshrined in the current EU regulatory framework for GMOs, is also suitable and appropriate for the assessment of genome-editing applications [16]. With a view to the requirements for a case-by-case ERA of a particular genome edited plant and its interaction with the receiving environment, general statements on hazards or risks may be even misleading. Considering the different characteristics of specific examples of genome edited plants and the respective risk issues relevant for these genome edited plants as described in Section 4, we believe that it is neither appropriate nor scientifically justified to draw general conclusions for whole groups of genome-editing applications, such as for all SDN-1/2 applications. Rather, a case-specific risk assessment approach, which is focused by a scientifically-based problem formulation to identify and address plausible risk issues, needs to be pursued [23]. We also note that if any conclusions on risks associated with a modified plant are based on the theoretical comparability of molecular modifications (or general “likeness”) this would constitute a major change to the current risk assessment approach for GMOs in the EU.

Secondly, the significant differences between naturally occurring mutations and targeted genetic modifications introduced by genome-editing tools, such as CRISPR/Cas or other SDN methods, need to be taken into account when assessing the assumed “likeness” of products obtained by genome-editing on the one hand and conventional breeding approaches on the other. It is not only important to consider which type of mutation may be introduced by either of the two approaches as EFSA did [16]. It is also crucial to consider where mutations may be introduced in the genome and how likely they are introduced at particular genomic loci. With a view to the significant differences concerning the relative frequencies of mutational events at specific genomic locations, we argue that a general “likeness” between genome-editing approaches and conventional breeding techniques cannot be assumed until such claims are supported by ERA results. We believe that genome editing and conventional breeding are two fundamentally different approaches. The first approach is directed to certain genomic sequences, whereas the latter is selecting from a pool of untargeted mutations, which display a certain bias concerning their genomic locations [31,32].

Only the most recent statement by the EFSA [21] recognises that a substantial number of genome-editing applications may result in the expression of many novel proteins and may display a high level of genetic and phenotypic complexity. The “depth of intervention” may be significantly higher for these genome-edited plants than for most conventional breeding products (see Section 3.2 and Section 4.2). While plants with complex modifications may present fewer challenges regarding detection and identification than genome-edited plants with single or few small-sized modifications, they will likely be more challenging for risk assessment. Overall, the conclusions on risk profiles for genome-edited plants should therefore not be based solely on plants harbouring only single-gene knockouts.

The lack of scientific knowledge regarding genome-edited plants with novel and complex traits, which are associated with fundamental changes in physiology and phenotype, needs to be considered as well. For this reason, plants with novel traits are regulated in Canada in case sufficient experience indicating a history of safe use is not available from practical use in an agricultural context [7]. “Familiarity” and “history of safe use” were also proposed as criteria by the EFSA for the risk assessment of plants produced with targeted mutagenesis, cisgenesis, and intragenesis [21]. However, at present it is unclear whether these proposals relate to the characteristics of specific plant varieties or to the wider range of plant species within the breeders’ gene pool and how these ideas can be based on specific comparisons to ensure a meaningful outcome. A legal analysis of the concept of “history of safe use” as applied in Directive 2001/18/EC and the ruling by the Court of Justice of the European Union on genome-editing applications conclude that the ”history of safe use” has to be applied in a very specific way and cannot be used as a general argument to exclude genome-editing applications from risk assessment [86].

In addition, the differences between unintended modifications generated with either genome-editing approaches or conventional breeding methods need to be addressed further, as described in Section 3.3. Looking only at the overall number of mutations introduced, without considering where they will occur and with which likelihood, is not appropriate for risk assessment. Therefore, we believe that the conclusions provided by the EFSA concerning the comparability of unintended modifications introduced with classical mutagenesis and genome-editing approaches, respectively [16], need to be reconsidered. Furthermore, we believe that there is no scientific justification to generally discount the relevance of unintended modifications created with genome editing with regard to risk assessment (see Section 5 for details). The current level of predictability of unintended modifications is not sufficient to preclude their assessment [23,48].

Another crucial issue is the higher speed of technical development in genome-edited plants compared with conventional approaches, such as classic mutagenesis. As indicated in Section 3.4, the speed of development is expected to be 2–4 fold higher, particularly if elite lines and commercially used varieties are directly edited, as indicated by Pixley et al. [50] and the EC study on NGTs [13]. Further, it needs to be considered that a higher speed of technical development will shorten the observational time to detect unexpected genotypic or phenotypic effects during product development with potential consequences for human and animal health and the environment. In our opinion, this is not sufficiently considered in the current discussion at the EU level.

## 7. Conclusions

In conclusion, we argue that a case-specific risk assessment approach is needed for genome-edited plants, based on the characteristics of the individual applications, taking into account their use and interaction with the receiving environment. Relevant for such an approach is the nature of the respective trait(s) developed in a particular genome-edited plant. A robust risk assessment approach is required in particular for genome-edited plants with novel and/or complex traits as well as traits with limited existing experience from practical use. Furthermore, a robust assessment of unintended modifications and possible adverse consequences of such modification needs to be maintained, based on an adequate molecular characterization of genome-edited plants and an appropriate comparative assessment.

On the way forward, several challenges have to be mastered, though. Further specific guidance for the ERA of genome-edited plants needs to be developed, as noted in the recent EFSA statement published in October 2022. However, the criteria for risk assessment proposed by the EFSA in this statement must be considered as the beginning of a discussion process rather than as an already suitable solution. The previous EFSA opinions on genome-editing applications, which were not aimed to establish specific guidance, are insufficient for that task, in our opinion. Appropriate data requirements for an ERA need to be established, taking into consideration available scientific data and existing experience for cultivated plants with similar properties. However, a robust molecular and phenotypic assessment is needed for plants with novel and/or complex traits. The basic data set currently required for GMOs could serve as a minimum requirement for such applications, with additional testing of possible environmental and/or food safety effects according to the characteristics for plants with novel and/or complex traits.

From a risk assessment viewpoint, generalizing conclusions on the assumed safety of broad groups of genome-editing applications and their “likeness” with conventional breeding approaches are not warranted using current scientific evidence. With a view to the existing uncertainties, rather a broad, but flexible risk assessment approach according to the case-by-case principle needs to be implemented to sufficiently address relevant biosafety issues of genome-edited plants. For plausible hazards associated with NGT and, in particular, genome-editing applications, the level of robustness in the evidence currently required for the ERA of GMOs needs to be maintained.

## Figures and Tables

**Table 1 plants-12-01764-t001:** Distribution of simple vs. complex SDN-1 applications in plants (published between 1996 and July 2019) for different trait categories according to the data of Menz et al. [34] and Kawall [35]. Indicated are percentages of single-gene knockouts vs. complex genomic modifications, including simultaneous editing of multiple gene copies present in a plant and multiplexed editing of different genes. Absolute numbers of applications are provided in parenthesis.

Trait Category	Single-Gene Knockouts	Complex Modifications
Enhanced breeding	85% (17)	15% (3)
Herbicide resistance	66% (2)	33% (1) ^1^
Agronomic value	57% (35)	43% (26)
Biotic stress resistance	53% (15)	47% (13)
Food and feed quality	51% (24)	49% (23)
Abiotic stress resistance	50% (3)	50% (3)
Industrial utilization	13% (1)	87% (7)

^1^ Note that only data for SDN-1 applications are included. Applications from additional sources covering other genome-editing types (e.g., SDN-2/3) were not included to maintain data consistency across trait categories.

## Data Availability

Not applicable.

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
