# Peer review of "Recommendations for the Assessment of Potential Environmental Effects of Genome-Editing Applications in Plants in the EU"

_plants, 2023, doi:10.3390/plants12091764_

Round 1

Reviewer 1 Report

This review article provides a thorough analysis of the current initiative by the EC on the use of new genomic techniques, particularly targeted mutagenesis and cisgenesis, in plant breeding. The authors highlight the importance of maintaining a high level of protection for human and animal health, as well as the environment, when using these techniques, and emphasize the need for a case-by-case approach to risk assessment. The article also discusses the limitations of the available EFSA guidance on conducting an environmental risk assessment (ERA) for genetically modified products and proposes essential elements for a risk-oriented regulatory approach.

Overall, this is a comprehensive and timely review that will be of interest to researchers working in the field of genome editing. The article covers all the important aspects of this field and provides a concise summary of recent advances. I believe this review is suitable for publication, and I commend the authors for their valuable contribution to the ongoing discussion on the regulation of genome-edited plants.

Author Response

Dear Reviewer1

Kind thanks for providing a review of our manuscript and for your overall recommendation that the manuscript is suitable for publication.

According to suggestions of the second reviewer a few editorial changes have been made to the text (e.g. adding long form names where the respective abbreviations were not introduced previously and explaining the SDN-1, SDN-2 and ODM approaches in a footnote). However these revisions do not change the substance of the manuscript in any way and therefore would not impact your current evaluation.

Many thanks again for your time and effort – very much appreciated!

Reviewer 2 Report

The manuscript submitted by Eckerstorfer et al. describes the current EU biosafety regulation for ERA of GM products. The authors proposed some key points for a case-by-case approach to the risk-oriented regulatory frame. The topic described would be interesting to a wide range of readers. I have found several issues as follows that, once addressed, I think that the manuscript will be improved.

Most readers may not be familiar with SDN-1/2 or ODM. The author should briefly explain what they are.

Minor comments

Line 30, provide the correct name of EFSA, without abbreviation.

Line 146, provide the correct name of ODM.

Lines 222 and 227, underlines should be removed from ‘and’.

Author Response

Dear Reviewer2

Many kind thanks for providing a review of our manuscript and for your suggestions for improvement, which were all addressed during revision (see below in response to your comments):

Most readers may not be familiar with SDN-1/2 or ODM. The author should briefly explain what they are.

Thank you for your observation! The specific techniques are mentioned for the first time in our text in a direct quote, therefore an explanation - as suggested by your comment - is provided in footnotes for the respective terms. This should be in line with the explanation for the term SDN, which was also provided as a footnote in the original manuscript.

Minor comments

Line 30, provide the correct name of EFSA, without abbreviation.

Thank you for your observation! The full name of the European Food Safety Authority is now provided in the text (L 30).

Line 146, provide the correct name of ODM.

The non-abbreviated name of ODM is now provided in the text (L 147), together with a brief explanation as suggested in your general comment.

Lines 222 and 227, underlines should be removed from ‘and’.

The underlines have been removed as suggested. Kind thanks for your observation!